# Amyloidogenic and Neuroinflammatory Molecular Pathways Are Contrasted Using Menaquinone 4 (MK4) and Reduced Menaquinone 7 (MK7R) in Association with Increased DNA Methylation in SK-N-BE Neuroblastoma Cell Line

**DOI:** 10.3390/cells13010058

**Published:** 2023-12-27

**Authors:** Michela Orticello, Rosaria A. Cavallaro, Daniele Antinori, Tiziana Raia, Marco Lucarelli, Andrea Fuso

**Affiliations:** 1Department of Experimental Medicine, Sapienza University of Rome, 00161 Rome, Italydaniele.antinori@uniroma1.it (D.A.); tiziana.raia@uniroma1.it (T.R.); marco.lucarelli@uniroma1.it (M.L.); 2Department of Surgery, Sapienza University of Rome, 00161 Rome, Italy; rosaria.cavallaro@uniroma1.it; 3Pasteur Institute, Cenci Bolognetti Foundation, Sapienza University of Rome, 00161 Rome, Italy

**Keywords:** DNA methylation, gene expression regulation, vitamin K2, menaquinone derivatives, MK4, reduced MK7, neurodegeneration, neuroinflammation, Alzheimer’s disease, prevention

## Abstract

Besides its role in coagulation, vitamin K seems to be involved in various other mechanisms, including inflammation and age-related diseases, also at the level of gene expression. This work examined the roles of two vitamin K2 (menaquinones) vitamers, namely, menaquinone-4 (MK4) and reduced menaquinone-7 (MK7R), as gene modulator compounds, as well as their potential role in the epigenetic regulation of genes involved in amyloidogenesis and neuroinflammation. The SK-N-BE human neuroblastoma cells provided a “first-line” model for screening the neuroinflammatory and neurodegenerative molecular pathways. MK7R, being a new vitamin K form, was first tested in terms of solubilization, uptake and cell viability, together with MK4 as an endogenous control. We assessed the expression of key factors in amyloidogenesis and neuroinflammation, observing that the MK7R treatment was associated with the downregulation of neurodegeneration- (*PSEN1* and *BACE1*) and neuroinflammation- (*IL-1β* and *IL-6*) associated genes, whereas genes retaining protective roles toward amiloidogenesis were upregulated (*ADAM10* and *ADAM17*). By profiling the DNA methylation patterns of genes known to be epigenetically regulated, we observed a correlation between hypermethylation and the downregulation of *PSEN1*, *IL-1β* and *IL-6.* These results suggest a possible role of MK7R in the treatment of cognitive impairment, giving a possible base for further preclinical experiments in animal models of neurodegenerative disease.

## 1. Introduction

Since its discovery in 1936 [1], vitamin K, which is synthesized by photosynthetic organisms or bacteria and metabolized mainly in the liver, has been recognized as a key cofactor for the modulation of the activity of blood-clotting factors. In the last decade, the bulk of research on vitamin K has shifted beyond coagulation, additionally showing its involvement in a wide range of biological processes. Vitamin K seems to exert an anti-apoptotic action through a pro-survival effect on brain cells [2] and in vitro studies showed that it is able to protect against oxidative damage and inflammatory cascade activation [3,4]. The reduced form of vitamin K acts as an antioxidant agent by protecting phospholipid membranes from peroxidation by direct reactive oxygen species (ROS) scavenging [5,6]. Recently, a study aimed at evaluating the effect of vitamin K on the redox metabolism of human osteoblasts cultured in the presence of hydroxyapatite-based biomaterials showed that vitamin K prevents redox imbalance by decreasing ROS levels [7]. Moreover, a currently active field of research around vitamin K relates to its potential anti-cancer effect. Indeed, vitamin K has been implicated in the growth inhibition of several neoplastic cell lines, mainly by inducing apoptosis and cell cycle arrest through various mechanisms [8]. Studies conducted in vitro and in vivo explored novel vitamin K roles in cardiovascular disease (CVD), along with brain and bone health [9,10]. In relation to bone health, the effect of vitamin K supplementation was evaluated in several clinical trials with results pointing to a protective effect through the improvement of bone quality, with increased strength, reduced turnover and a reduction in fractures [11]. In relation to brain health, vitamin K induces the synthesis of sphingolipids, which are amphipathic lipids that are an essential part of the central nervous system (CNS) cell membrane for neuronal proliferation and differentiation [12]. Sphingolipids guide the process of myelination in the CNS and are themselves major components of the oligodendrocyte membrane. In addition, they have been related to neuro-inflammation and neurodegeneration due to microglial activation and the accumulation of amyloid precursor protein (APP) [13]. Moreover, vitamin K is involved in the biological activation of the vitamin K-dependent proteins S and Gas6, which are important growth and cell survival factors in the brain [14,15,16,17]. Recent evidence showed the role of vitamin K in Alzheimer’s disease (AD), considering the physiological role in aging and underlining the beneficial effects for brain health [18,19,20,21]. Together with clinical studies pinpointing the beneficial effect of vitamin K on a plethora of biological processes, the possible toxicity of vitamin K was also investigated, concluding that there are almost no reported cases of natural vitamin K systemic toxicity, either in animals or in humans. For instance, no hypercoagulable state was observed in individuals consuming doses above the recommended daily allowance of 75 micrograms of vitamin K (Commission Directive 2008/100/EC) [22]. Likewise, cases of extremely high levels of vitamin K intake were reported without adverse effects [23,24]. The absence of adverse effects or documented toxicity strongly sustains the use of vitamin K as a health-promoting supplement. In fact, a major economic challenge for healthcare systems is caused by aging societies; therefore, diet supplements promoting healthy aging and improving the prognosis of age-related diseases, are required to be implemented in clinical practice.

Vitamin K occurs in two forms, the phylloquinone vitamin K1, along with vitamin K2, which includes a range of vitamers referred to as menaquinone-n (MK-n) forms, where the “n” reflects the number of repeating 5-carbon units (Figure 1). Most K2 vitamers are produced by bacteria, whereas MK4 can be found in fish, eggs, ground beef, milk, butter, and fermented cheese or vegetables. Some vitamin K2 vitamers, such as MK7, MK8 and MK-9, which are recognized as the main forms in terms of nutrition value [25], are biosynthesized by several obligate and facultative anaerobic bacteria [26,27].

In terms of chemical structure, K2 vitamers are characterized by the presence of a common 2-methlyl-1,4-naphtoquinone ring and a hydrophobic poly-isoprenoid side. Properties and activities depend on the lipophilic side chain and the aromatic ring. These vitamers can more easily cross the blood–brain barrier thanks to their lipophilic nature [28,29]. The side chains vary in length and degree of saturation, and their impacts on the fat solubility of each vitamer [30,31] explain the differences in terms of absorption, distribution and bioavailability. For instance, the absorption rates of vitamin K derivatives decrease significantly with the length of the side chain [32]. Once vitamin K is incorporated into the rough endoplasmic reticulum of the cells, it is reduced into the biologically active form hydroquinone (or KH2). This form acts as a cofactor of the γ-glutamyl carboxylase (GGCX enzyme) and is converted into vitamin K-2,3 epoxide (or KO); then reduced into the respective quinone by vitamin K epoxide reductase complex subunit 1 (VKORC1); and finally, it is transformed back into hydroquinone by the vitamin K reductase. Then, the cycle starts again. Various drugs, such as coumadin (i.e., warfarin), inhibit vitamin K oxide reductase [33]. In the absence of vitamin K, or with a blockade of VKORC-1 by coumadin anticoagulants, vitamin K quinone can be reduced, at least in the liver, to vitamin K hydroquinone by a cytosolic NAD(P)H–dependent oxidoreductase, but vitamin K epoxide is not a substrate for this enzyme, and thus, vitamin K recycling is not enabled (Appendix A). Therefore, under coumadin treatment, patients are not able to metabolize and use vitamin K derived from nutrition, which results in vitamin deficiency.

At the very beginning, we focused our attention on a synthetic K2 reduced form, namely, MK7R, as a formulation that could potentially allow coumadin-treated patients who have their vitamin K cycle compromised to enjoy the beneficial effects of such a powerful antioxidant. In fact, the reduced MK7 form could bypass the reductase enzyme activity that is inhibited under coumadin or warfarin-based treatment [34].

Even if both vitamin K1 and vitamin K2 play major biological roles in the brain, vitamin K2 (mainly MK4) is the most abundant in the brain, leading to research related to AD being more prone to use vitamin K2 [17]. Moreover, vitamin K2, particularly the MK7 vitamer, has a higher bioavailability and a longer circulation half-life compared with vitamin K1, and thus, it is more easily involved in extrahepatic tissue reactions [35,36]. Additionally, diet supplementation with MK7 is able to increase the MK4 levels in extrahepatic tissues [37].

On the other hand, diet has recently been receiving particular attention with respect to its ability to exert indirect or direct epigenetic activity [38]. Epigenetic modifications, consisting of reversible heritable changes in gene expression without DNA sequence alteration and including DNA methylation, histone modifications and chromatin remodeling [39], play an important role in disease pathogenesis. In particular, the degree of DNA methylation defines gene expression [40].

Several external, as well as internal, factors regulate the epigenetic machinery, either directly or indirectly [41]. Hence, a growing body of evidence suggests vitamins as major epigenetic modifiers attracting the attention of consumers, nutritionists and scientists, along with several other vitamins, including vitamins A, B, C and D, that can regulate the epigenetic machinery [42,43,44].

Based on these considerations, we decided to investigate the efficacy of some vitamin K2 derivatives in terms of contrasting or modulating neurodegenerative and age-related features. Specifically, we used the neuroblastoma cell line for a preliminary screening aimed at evidencing the possible modulation of key factors involved in neurodegeneration and neuroinflammation. Therefore, we investigated the expression of genes involved in the amyloidogenic and non-amyloidogenic pathways, as well as pro-inflammatory cytokines, using real-time PCR, Western blotting and ELISA assays. Moreover, since some of the genes that were modulated by vitamin K are known to be epigenetically modulated by DNA methylation, we also profiled the CpG and non-CpG methylation status of their promoter regions.

For this study we used MK4, MK7, MK7R and diacetylated-MK7, with the latter being a tentative contender to overcome the easy oxidation of MK7R, and thus, a possible way to guarantee the effective delivery of MK7R into cells. Based on preliminary assays and screenings on vitamin K synthetic compounds, we selected the two most stable and soluble menaquinone derivatives to be administered to neuroblastoma cells in culture, namely, MK4 and MK7R (Figure 1). MK7 and diacetylated-MK7 were discharged after the initial screening due to their poor solubility, which could not guarantee the necessary reliability in cell culture experiments.

## 2. Materials and Methods

### 2.1. Vitamin K Derivatives

The vitamin K2 vitamers (MK4 and reduced MK7R) were all provided by Gnosis by Lesaffre (Desio, MB, Italy). The reduced form of MK7 was prepared by Gnosis by Lesaffre following a patented procedure among the ones described in the patent US10563270. Briefly, menaquinone 7 was reduced with sodium dithionite in ethyl acetate under nitrogen flux and vigorous stirring. At the end of the reaction, the mass was concentrated under vacuum, filtered and dried. The crystalline product, which was maintained under vacuum, was found to be stable, retaining 98% of the initial reduced form content after 3 months of storage. In the present experiment, MK7R was freshly prepared; kept under vacuum until its use; and, once opened, immediately added to the cell culture.

Due to their poor solubility in water, the vitamin K2 vitamers were dissolved in EtOH or DMSO in a first attempt and tested for preliminary effects (cell growth and viability). Since they showed similar solubility, we decided to use only MK4 (as a control) and MK7R in EtOH for the supplementation in the cell culture media.

### 2.2. Cell Culture

SK-N-BE human neuroblastoma cells were grown in Dulbecco’s Modified Eagle Medium (DMEM)/high glucose supplemented with 10% fetal bovine serum (FBS), 1% L-Glutamine and antibiotics (Penicillin 100 IU/mL, Streptomycin 100 μg/mL) (all from; Euroclone, Milan, Italy) at 37 °C, 5% CO_2_ and 80% humidity.

To evaluate the effect on cellular viability, different vitamin concentrations in cell culture medium were tested (0, 2, 5, 10, 20, 50, 100 μM) on the basis of previous reports [19,45,46,47,48,49]. Cell cultures were stopped via trypsinization every 24 h for six days to evaluate the cell morphology (before the stop) and the growth rate via cell counting (Appendix A). No significant alterations were observed with MK4 and MK7R.

The 50 μM MK4 and MK7R conditions were chosen for the further assessment of gene expression on the basis of the growth curve. To this end, and according to the experimental plan, human neuroblastoma SK-N-BE cells were seeded in complete DMEM/high glucose; after 24 h of growth (T0), the cells were shifted to DMEM supplemented with 50 µM of vitamin K2 vitamers dissolved in EtOH (1 μL EtOH per 1 mL DMEM). Cells were treated with the vitamers for 48 h for gene expression and DNA methylation analysis or for 72 h for protein analysis. Fresh media was used to replace the old media every 48 h. Control media was added with 1 μL EtOH per 1 mL DMEM.

### 2.3. UPLC-PDA Chromatography for Vitamin K Determination

The UPLC-PDA consisted of a Waters ACQUITY UPLC system (Waters, Milford, MA, USA), including a quaternary solvent manager (QSM), a sample manager with flow through needle system (FTN) and a photodiode array detector (PDA). For the determination of the MK4, MK7 and MK7R in cell lysates, we modified a previously described UPLC-PDA chromatographic method [50] that is able to separate all the vitamers well in the same chromatographic run. This allowed us to follow the effect of the supplementation on the level of vitamin K2 vitamers, together with the eventual transformation and the oxidation of the MK7R to MK7, as well as monitor the uptake by the cells. Chromatographic analyses were performed on a Phenomenex Kinetex C18 100A, 2.6 μm (4.6 × 100 mm) column associated with a similar guard column, thermostated at 25 °C. Isocratic elution was performed with 95% methanol (solvent A) and 5% H_2_O (solvent B) at a flow rate of 0.8 mL/min for 15 min and monitoring via UV spectra comparison using a PDA detector reading at 254 nm (245 nm for reduced MK7R). SK-NB-E cells treated with the different vitamin K vitamers for 48 h were then lysed in PBS. The cell lysates were denatured in EtOH, extracted using n-hexane, collected and evaporated under nitrogen. The residue was reconstituted with 150 μL of ethanol, filtered through 0.2 μm Sartorius filters and subjected to UPLC analysis using the method mentioned above. The identification of the different vitamers was made by using standard analytical compounds. The instrument control and data acquisition were carried out using Waters Empower version 3 software (Waters, Milford, MA, USA).

### 2.4. Quantitative Real-Time PCR Analysis

RNA purification was performed using the RNeasy kit (Qiagen, Milan, Italy), following the instructions indicated by the manufacturer, and the cDNA was synthesized via retro-transcription from 1 µg of total RNA. A total of 0.5 µL of cDNA was used for the real-time reactions. Analyses were performed in triplicate for each sample. cDNA levels were normalized to the β-actin control and presented as the fold increase over the control sample, as previously described [51]. The RNA extraction on columns was carried out using the automated tool QIAcube (Qiagen, Milan, Italy). The extracted RNA was resuspended in 30 µL.

Purified mRNA was reverse transcribed back into cDNA using the OptiFast cDNA Synthesis (Optiprime, Aurogene, Roma, Italy) according to manufacturer instructions.

The real-time analysis for assessing the expression of target genes was performed on a CFX apparatus using the iTaq Universal SYBR Green Supermix (both from Bio-Rad, Milan, Italy), as previously described [51]. Sequences and characteristics of the primers used in the PCR reactions are reported in Table 1. The results were analyzed with the CFX Connect (CFX Maestro version 2.3, Bio-Rad, Milan, Italy) instrument software using ANOVA. The results were considered significant for *p* < 0.05.

### 2.5. Protein Analysis

SK-NB-E cells were treated for 72 h with 50 µM of MK4 and MK7R and then lysed for 30 min on ice in RIPA buffer (50 mM Tris, 150 mM NaCl, 1 mM EGTA, 1 mM EDTA, 1 mM sodium deoxycholate, 1% NP40), supplemented with PMSF (200 μM), leupeptin (1 μM), pepstatin A (1 μM) and calpain inhibitor I (5 μM), all from Merck (Milan, Italy). A total of 60 μg of each protein extract was used for electrophoresis with NuPAGE Bis-Tris precasted 10% or 7% (based on the molecular weight under investigation), run in NuPAGE MOPS buffer (Invitrogen, Milan, Italy) under reducing conditions and transferred onto nitrocellulose membranes, as previously described [51]. The primary antibodies used are listed in the table below (Table 2). Antigens on the membrane were revealed using enhanced chemiluminescence (ECL plus, Amersham, UK). Densitometric analysis of immunoblots was performed using Canonscan Toolbox 10.5 software (Canon, Cernusco sul Naviglio, Italy) for the image acquisition and Quantity One 1D software (Bio-Rad, Milan, Italy) for the analysis. Optical densities (O.D.s) from at least three different experiments were calculated for each sample and normalized with the corresponding 14,3,3β signal O.D., which was chosen since it was unmodulated in these cells in many different experimental setups [51].

### 2.6. Enzyme-Linked Immunosorbent Assay (ELISA)

ELISA tests were used according to the instructions of the manufacturer to determine the levels of the cytokines secreted in the cell culture medium. The following ELISA kits were used: IL-1β, sensitivity—6.5 pg/mL (Abcam, ab46052, Cambridge, UK); IL-6, sensitivity—0.7 pg/mL (Quantikine, R&D, D6050, Milan, Italy). The results of the ELISA tests were assessed using an Opsys MR microtiter plate reader (Dynex Technologies, Chantilly, VA, USA).

### 2.7. DNA Methylation Assay to Profile CpG and Non-CpG Methylation

The CpG and non-CpG DNA methylation profile of PSEN1, IL-1β and IL-6 5′ flanking was assessed via bisulphite DNA modification and genomic sequencing using non-CpG methylation-insensitive primers (MIPs), as previously described [52,53,54]. MIPs amplify the target DNA sequences with unpredicted non-CpG methylation in an unbiased way. Briefly, DNA was purified from cultured cells through the DNeasy Blood and Tissue Kit (Qiagen, Milan, Italy) and the Qiacube. The bisulphite treatment was performed using the EpiTect Bisulphite kit (Qiagen, Milan, Italy). The PCR products were cloned in the pCR2.1 vectors using the TA Cloning Kit (Thermo Fisher, Milan, Italy). At least 20 clones for each experimental condition were analyzed. Sequencing was performed via the cycle-sequencing method using the ABI PRISM 3130xl genetic analyzer (Thermo Fisher, Milan, Italy) and the M13 primers. The methylation status of any cytosine (both CpG and non-CpG) in each sequenced clone was annotated. For each experimental sample, the methylation percentage of any single cytosine was calculated as the number of methylated cytosines divided by the number of sequenced clones × 100 [55].

The characteristics of the MIPs used for bisulphite analysis were previously described [54]. The primers used allowed us to assess the methylation status of the plus (5′- > 3′) DNA strand. Positive and negative controls, which were necessary to check the conversion efficiency, were performed as previously described [52,53,54].

### 2.8. Statistical Analysis

At least three independent biological experiments (*N* = 3) were performed and three technical replicates of each experiment were analyzed for each assay. One-way ANOVA and Tukey’s post-test were used for mRNA and protein expression evaluation. Contingency tables and chi-square and Fisher’s exact tests were applied for the DNA methylation results. All bar plots show the mean value ± standard deviation. Asterisks in the figures indicate statistically significant differences. All the statistical analyses were computed using SPSS software V27.

## 3. Results

### 3.1. Solubility and Uptake

The very first challenge in testing vitamin K vitamers as potential supplements was the solubility assessment. Vitamin K is a lipid-soluble compound and it is not hydrophilic; therefore, we first tried to dissolve the vitamin K vitamers either in EtOH or DMSO. MK7 and diacetylated MK7 showed very poor solubility and scarce reliability in the cell growth analysis. Therefore, we decided to perform the rest of the assays using MK4 and MK7R dissolved in EtOH. To evaluate the cell morphology and growth rate, it was useful to set up the working concentrations; the human neuroblastoma cells (SK-N-BE cell line) were treated with the MK4- or MK7R-supplemented medium in concentrations from 0 up to 100 µM, and then stopped and counted every 24 h for six days. No significant alterations in viability were observed in the treated cells after 144 h for both the MK4 and MK7R treatments with respect to the untreated control (Appendix A). The 50 µM concentration was then used for the further assessment of DNA methylation and gene expression. For the protein analysis and the uptake assessment, only the 50 µM concentration for both vitamers, which showed the highest effect, was considered.

In parallel with the solubility test and the cell viability assay, we set up a method to study the vitamers uptake in the treated cells. For the determination of the two soluble vitamers, namely, MK4 and MK7R, and the corresponding oxidized MK7 in cell lysates, we used a UPLC-PDA chromatographic method [50], which is able to separate all the vitamers well in the same chromatographic run, following the instructions of the molecule provider (Gnosis by Lesaffre). This allowed us to assess the uptake and the transformation (oxidation of MK7R to MK7) of the vitamers after supplementation and was not intended as a quantitative analysis. The SK-N-BE cells treated with MK4 or MK7R 50 μM for 48 h showed that there was an increase in MK4 levels in the MK4-treated cells. In contrast, in the MK7R-treated cells, both the oxidized and reduced MK7 peaks were detectable at 2 min and 9 min, respectively, compared with the untreated sample. This was expected based on the MK7R oxidation to MK7. Thus, the chromatogram showed two peaks in cells treated with the MK7R, meaning that MK7R was taken but mostly converted into the oxidized MK7 form detectable at 9 min (Appendix A). Given the very poor solubility of MK7, the supplementation of the more soluble MK7R, which is rapidly oxidized into MK7, can represent an easier way to study the effect of MK7 in a cell culture.

### 3.2. Neurodegeneration-Associated Pathway

MK4 and MK7R at the concentration of 50 µM modulated the expressions of some key genes involved in the amyloid pathology, such as *PSEN1*, *BACE1*, *APP*, *ADAM17* (TACE) and *ADAM10* (Figure 2). The APP gene, which encodes for the APP that serves as a substrate for the β- and γ-secretases, was not significantly modulated in the cells treated with MK4 and MK7R at the concentration of 50 µM with respect to the APP gene expression level in the control cells (Figure 2a). However, *BACE1* and *PSEN1* (which respectively encode for the active sites of the β- and γ-secretases) transcript levels decreased to approximately 0.6-fold in neuroblastoma cells treated with either MK4 or MK7R with respect to control (Figure 2b,c; *F* = 5.09, *p* < 0.05 MK4 vs. ctrl; *F* = 5.4, *p* < 0.01 MK7R vs. ctrl). The other genes considered were the genes that encode for the enzymes ADAM17 and ADAM10, having α-secretase activity on the APP and involved in the non-toxic pathway. Treatment with MK4 and MK7R 50 µM increased these gene expressions, reaching 1.9-fold within 48 h in the cells treated with MK7R (Figure 2d,e).

The modulation of the key factors observed at the mRNA level was also investigated using Western blotting analysis at the protein level for the genes reported as significantly deregulated (Figure 3). Original WB images are shown in Appendix A. The analysis was performed on cells treated with 50 µM MK4 and MK7R; the figure reports representative WB images on the left (Figure 3a) and the bar plots obtained using triplicate analysis (on three different cell experiments) on the right (Figure 3b–e). The BACE1 enzyme, which is involved in the amyloidogenic pathway, was not significantly modulated in response to MK4 (Figure 3b), whilst it decreased to 0.6-fold in response to MK7R (*F* = 5.6, *p* < 0.01). We also observed a decrease in the PSEN1 protein levels when cells were treated with MK4 or MK7R (Figure 3c; *F* = 4.3, *p* < 0.05). As for the non-amyloidogenic pathway, increased secretase activity exerted by ADAM17 on APP, as suggested by the increase in sAPPα peptide (Figure 3c), was observed only when the cells were treated with MK7R (*F* = 5.2, *p* < 0.01). Among the two α-secretases, only ADAM17 was modulated by vitamin K at the protein level, showing upregulation in the cells treated with MK7R (Figure 3d; *F* = 4.5, *p* < 0.05). ADAM10 was not modulated in WB (Figure 3f).

### 3.3. Neuroinflammation-Associated Pathway

In the attempt to assess the effects of MK4 and MK7R on neuroinflammatory molecular pathways in SK-N-BE cells, we investigated the expression of two relevant pro-inflammatory cytokines, namely, IL-1β and IL-6.

The most significant reduction at the transcript level was observed for the gene encoding *IL-1β*, whose expression halved within 48 h in response to the administration of 50 µM MK4 or MK7R (Figure 4a; *F* = 5.08, *p* < 0.01). A significant decrease in *IL-6* was also observed with either MK4 or MK7R (Figure 4b; *F* = 4.6, *p* < 0.05).

Regarding the effects of mRNA modulation on cytokine secretion, we observed that MK4 and MK7R both promoted the reduction in the circulating pro-inflammatory cytokines IL-1β (Figure 4c; *F* = 4.8, *p* < 0.01) and IL-6 (Figure 4d; *F* = 4.25, *p* < 0.05) at 72 h. Interestingly, NF-kB (Nuclear Factor-kB) was downregulated only in the MK7R-treated cells (Figure 4e,f; *p* < 0.05).

### 3.4. DNA Methylation Analysis

Since some of the genes investigated in this work, specifically *PSEN1*, *IL-1β* and *IL6*, are known to be epigenetically modulated by the DNA methylation of their 5′-flanking regions in different experimental settings [45,46], we decided to explore the possibility that vitamin K treatments could, although indirectly, induce epigenetic changes in these genes. We, therefore, assessed the methylation profile of the *PSEN1*, *IL-1*β and *IL-6* 5′-flanking region at the single-cytosine level through bisulfite modification of genomic DNA, followed by the cloning of PCR products into plasmids and Sanger sequencing. The use of the PCR MIPs, which were unbiased toward non-CpG methylation, allowed us to assess both CpG and non-CpG methylation percent [43,52].

The bar plots in Figure 5, Figure 6 and Figure 7 report the percent methylation of each CpG and non-CpG (CpA, CpT, CpC) cytosines in the 5′-flanking region of the *PSEN1* (from cytosine 974–1226; Figure 5), *IL-1*β (from cytosine 943–1203; Figure 6) and *IL-6* (from cytosine 1446–1648; Figure 6) genes. It was observed that all three sequences were hypermethylated in the MK4-treated cells vs. ctrl (Z = −1.6, *p* < 0.05) and even further hypermethylated in the MK7R-treated cells (Z = −2.4, *p* < 0.01 vs. ctrl; Z = −1.8, *p* < 0.05 vs. MK4-treated cells), in agreement with the observed reduced expression in the same experimental conditions.

In order to gain some further insight related to the methylation changes associated with the treatment, we analyzed the expression of the DNA methyltransferases (DNMT1, 3a and 3b) and observed no significant differences in these three genes in cells treated with MK4 and MK7R, although a trend toward upregulation was observed for DNMT3b (Appendix A).

## 4. Discussion

Today we know that the lipid-soluble compounds referred to as vitamin K have a plethora of beneficial effects. Recent studies suggest that aging and age-related diseases can benefit from novel approaches lying partially in diet and its modifications. The applications of nutrigenomics, therefore, represent a potential approach to prevent or reduce neurodegenerative risk through nutritional interventions. Growing evidence from the literature shows that vitamin K can impact health conditions beyond bone and cardiovascular health. The vitamin K family is involved in functions such as cell survival, chemotaxis, mitogenesis, cell growth and myelination mediated by the activation of vitamin K-dependent proteins [35]. The K2 vitamer known as MK7 was shown to have advantages given its superior bioavailability and higher half-life in circulation when compared with other K vitamers. A study shows that MK7 has a half-life time of 68 h compared with only 1–2 h for K1 [35]. This increased half-life results in more stable blood levels and higher bioavailability of MK7, and thus, is more able to reach extra-hepatic tissues [18,35,36,56]. Moreover, high levels of MK4 and MK7 have no documented toxicity or adverse health effects [22,23,24]. The most important function of vitamin K broadly investigated in research is the gamma-carboxylation of the glutamic acid residues and activation of vitamin K-dependent proteins (VKDPs). Additionally, recent findings suggest that vitamin K2 also displays other non-VKDP-dependent activities, notably because K2 vitamers seem capable of directly binding to the proteins, and thus, modulating their activity without affecting their gene expression [36,47]. Saputra and collaborators [48] showed that MK4 attenuates microglial inflammation by inhibiting NF-kB signaling in mouse-microglia-derived MG6 cells and by downregulating the expressions of IL-1β and IL-6.

Although recent studies underlined the protective role of vitamin K2 in neurodegenerative diseases, some mechanisms still need to be elucidated, in particular the promising role of vitamin K2 in AD [57,58]. It was demonstrated that there exists a vitamin K2 protective effect against Amyloid-beta (Aβ) cytotoxicity in neural cells via regulating the phosphatidylinositol 3-kinase (PI3K)-associated signaling pathway and inhibiting caspase-3-mediated apoptosis [49]. As a matter of fact, several studies showed the association between vitamin deficiency and neurodegenerative disease and underlined the protecting role of nutritional intervention to prevent or counteract neurodegenerative damage. Our previous studies evidenced the role of B vitamins as dietary factors connecting nutrition and epigenetic modulation of brain disease, showing that B vitamin deficiency is associated with the impairment of the methylation potential, causing DNA hypomethylation and the consequent overexpression of PSEN1, leading to Aβ accumulation and exacerbation of AD-like features in TgCRND8 mice [59]. Conversely, S-adenosylmethionine supplementation was able to counteract the exacerbation of AD-like features, preventing PSEN1 hypomethylation and overexpression, thus reducing β- and γ-secretase activities, Aβ production and amyloid plaques spreading to control-like values in TgCRND8 mice in a B-deficient condition [60].

In this work, we studied the actions of two vitamin K2 vitamers, namely, menaquinones MK4 and MK7R, in SK-N-BE cells by investigating their potential role in the regulation of specific target genes involved in amyloidogenesis (*PSEN1*, *BACE1*, *APP*, *ADAM10*, *ADAM17*) and neuroinflammation (*NFkB*, *IL-1β*, *IL-6*). While MK4 represents a well-established vitamin K vitamer, which was already used in basic and translational studies, MK7R represents a different formulation that should allow for increased bioavailability, uptake and effects. This idea as the basis of this present study was preliminarily confirmed by the chromatographic analysis by showing that despite rapid oxidation, the use of MK7R in the culture medium guaranteed increased uptake of the vitamin. Since this compound also displays higher solubility (comparable to MK4), we can speculate on the use of this specific derivative for further studies aimed at exploring its translatability. Also, from the molecular point of view, we observed, in general, a higher effect of MK7R with respect to MK4: both derivatives showed the same molecular effects on target factors with respect to the control conditions, but MK4 appeared to be less efficient than MK7R.

We report a significant reduction in the expression of PSEN1 and BACE1 indicating that at the concentration of 50 µM, the tested vitamers interfered with the amyloidogenic pathway. On the other hand, the gene expression analysis using RT-PCR demonstrated an upregulation of ADAM17 and ADAM10 genes involved in the non-toxic pathway. The enhanced activity of ADAM17 and ADAM10 leads to the increasing secretion of neuroprotective soluble APPα fragments and reduction in Aβ generation, which may be beneficial in AD [61,62]. At the protein level, the upregulation was evident only for ADAM17. This could be sufficient to sustain the non-amyloidogenic pathway according to the increased sAPPα peptide. In fact, we observed an increase in sAPPα peptide, which is the product of the secretase activity exerted by ADAM17 on APP, in contrast to the amyloidogenic APP processing exerted by BACE1 and PSEN1.

In parallel to the neurodegenerative pathway, we also investigated key factors of the neuroinflammatory pathway, which is known to be involved in, if not causative of, neurodegenerative diseases [63]. In our analysis, the neuroinflammatory pathway also appeared to be modulated by both MK4 and MK7R. The K vitamins, indeed, were able to reduce the inflammatory response via NF-kB signaling pathway inactivation, which, in turn, resulted in significantly suppressing the expression of IL-1β and IL-6 in neuroblastoma cells. The decrease in protein levels was consistent with the gene downregulation.

The remarkable gene expression modulation that was strongly sustained by a coherent protein modulation exerted by MK7R was a great input for us in order to investigate the potential epigenetic modifications indirectly promoted by MK7R and is responsible for the gene expression outcomes in the experimental conditions compared with the control conditions and MK4. We analyzed the DNA methylation profiles of *PSEN1*, *IL-1β* and *IL-6* since these three genes were previously reported to be modulated by the methylation of their 5′-flanking regions in SK-N-BE cells, although in different experimental settings and in the brains of AD subjects vs. healthy age-matched controls [53,54,55,64,65]. This background motivated the idea to investigate the DNA methylation in cells treated using vitamin K derivatives, even if the direct epigenetic effects of vitamin K are not known. The results confirmed that the “protective” role of vitamin K derivatives, which is exerted through the mitigation of the neurodegenerative and neuroinflammatory pathways, is mediated by (or at least associated with) the hypermethylation of the regulatory regions of *PSEN1*, *IL-1β* and *IL-6*. Vitamin K treatment is also associated with *BACE1*, *ADAM10* and *ADAM17* modulation, but we did not go further in assessing the methylation pattern of these genes since in our previous research, we found that even when these mRNAs were modulated in association with hypo- and hypermethylating treatments, their promoters showed conserved methylation [58,64]. This observation suggests that some still-unknown “mediator” is involved in the connection between methylation metabolism and the expression of these genes.

We cannot, at this level, speculate about the mechanisms by which vitamin K can induce the hypermethylation of target genes. It is probably to be ascribed to indirect effects, maybe mediated by the vitamin K-dependent γ-carboxylation of effector intermediates. This aspect is of course intriguing and deserving of further investigation in the future, probably through “omic” analysis at the mRNA and protein levels. Another interesting aspect of the methylation data is that, as previously showed, the presence of methyl groups at non-CpG cytosine moieties (the so-called “non-CpG”, or “CpH”, methylation) is detectable at discrete levels when analyzed through bisulfite primers that are unbiased versus CpA, CpC and CpT methylation [51]. Moreover, non-CpG methylation appears to also be functional since it is modulated according to gene expression. The mechanism by which vitamin K should impact DNA methylation is not yet known. We tried to see whether the altered expression of the DNA methyltransferases could be involved but observed no significant modulation of DNMT1, 3a or 3b. The slight trend toward the upregulation of DNMT3b could be informative of the non-CpG methylation increase, but the lack of statistical significance prevents any further speculation.

The major limit of this present work was the use of a tumor cell line that is not a proper model of neurodegeneration or neuroinflammation. However, the SK-N-BE cells represent a very useful preliminary (first-line) model to screen the molecular targets of exogenous compounds since these cells express all the genes involved in amyloid processing at discrete levels and show detectable basal levels of pro-inflammatory cytokines. The data collected on vitamin K-treated SK-N-BE cells, and presented here, will allow for refining the analyses that we plan to execute on animal models, such as the TgCRND8 AD mice, primary neurons derived from the same mice and iPSCs derived from patients.

Overall, the results obtained, although still preliminary, are encouraging and suggestive of a neuroprotective effect of MK4 and, particularly, of reduced MK7 vitamers (Figure 8).

## 5. Conclusions

Vitamins are defined as organic compounds that are required in small amounts for the development and normal functioning of the body. The body cannot synthesize vitamins, either at all or not in sufficient quantities. As such, they must be obtained through the diet. The typical balanced diet of a healthy population has a rich number of vitamins, which prevent several diseases [66,67]. Especially in infants and elderly people, vitamin deficiency is common [68], and there is evidence showing that prolonged deficiencies in vitamins lead to malnutrition and severe health issues [69,70]. This is because vitamins commonly function as antioxidants or enzymatic cofactors [71,72]. A deficiency in, or a dysregulation of, specific vitamins adversely affects neuronal metabolism, which may lead to neurodegenerative diseases [73,74,75,76]. This prompted researchers to explore the role of different vitamins in the development and progression of diseases.

However, the conclusion on the impacts of MK7R on the amyloidogenic and neuro-inflammatory processes needs further investigation. Therefore, we hypothesize that the outcomes of our study will help in reaching a clear consensus on the effects of vitamin K supplementation on age-related diseases affecting the brain.

## Figures and Tables

**Figure 1 cells-13-00058-f001:**
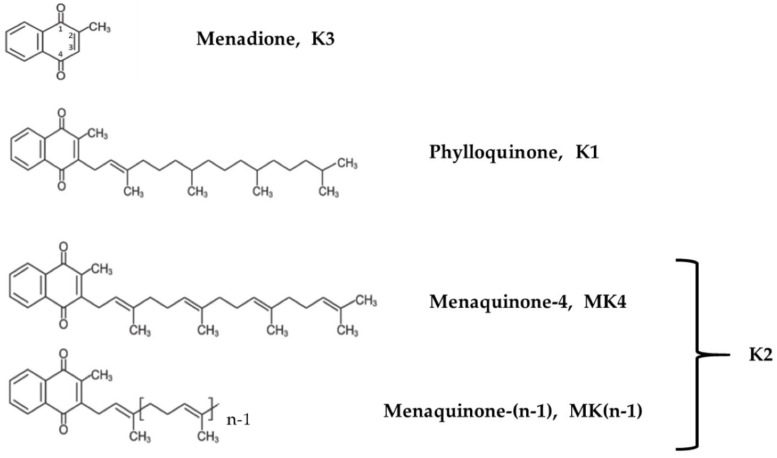
Chemical structures of vitamin K derivatives.

**Figure 2 cells-13-00058-f002:**
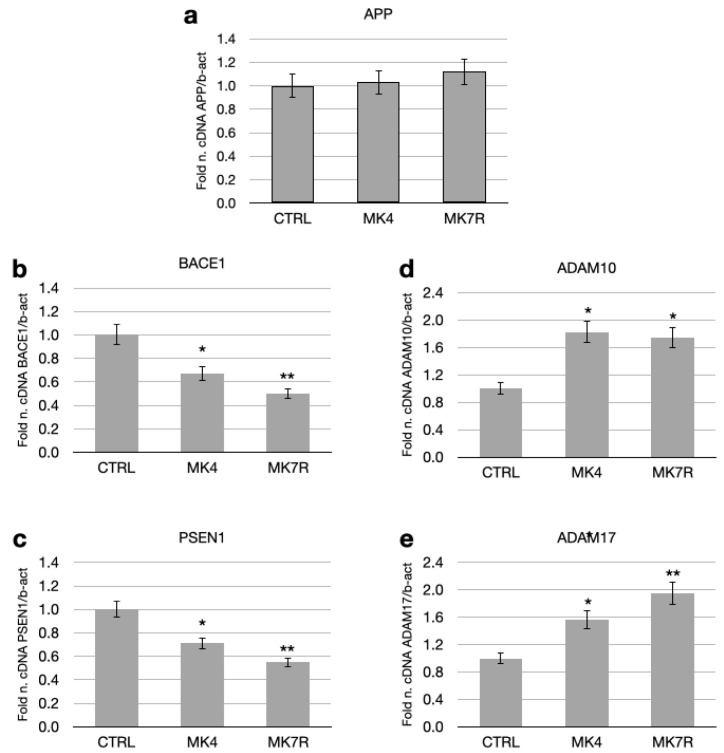
mRNA expressions of genes involved in neurodegenerative processes (amyloid processing) in SK-N-BE cells treated for 48 h with 50 µM MK4 or MK7R. (**a**) APP, (**b**) BACE1, (**c**) PSEN1, (**d**) ADAM10 and (**e**) ADAM17. Bar plots show the relative amounts of the target genes normalized to β-actin internal reference obtained using real-time PCR on the *y*-axis. Abbreviations: MK4, menaquinone 4; MK7R, reduced menaquinone 7. * *p* < 0.05, ** *p* < 0.01 vs. ctrl; *N* = 3.

**Figure 3 cells-13-00058-f003:**
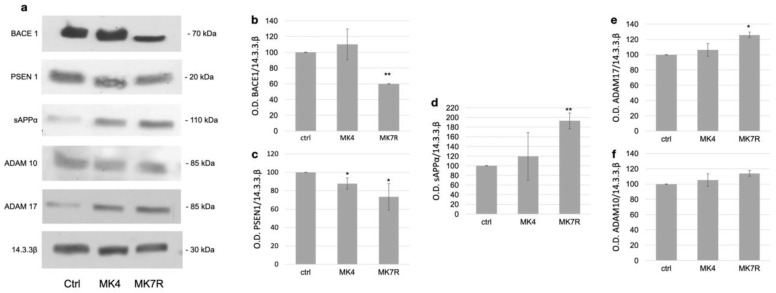
Western blotting analysis of markers of neurodegeneration in SK-N-BE cells treated for 72 h with 50 µM MK4 or MK7R. (**a**) Representative Western blot images of BACE1, PSEN1, sAPPα, ADAM17 and 14.3.3β proteins. (**b**–**f**) Densitometric analysis of protein bands obtained in at least 3 independent experiments using the 14.3.3β band as internal reference. Values are expressed as the percent of the control. (**b**) BACE1, (**c**) PSEN1, (**d**) sAPPα, (**e**) ADAM17 and (**f**) ADAM10. Abbreviations: MK4, menaquinone 4; MK7R, reduced menaquinone 7. * *p* < 0.05, ** *p* < 0.01 vs. ctrl; *N* = 3.

**Figure 4 cells-13-00058-f004:**
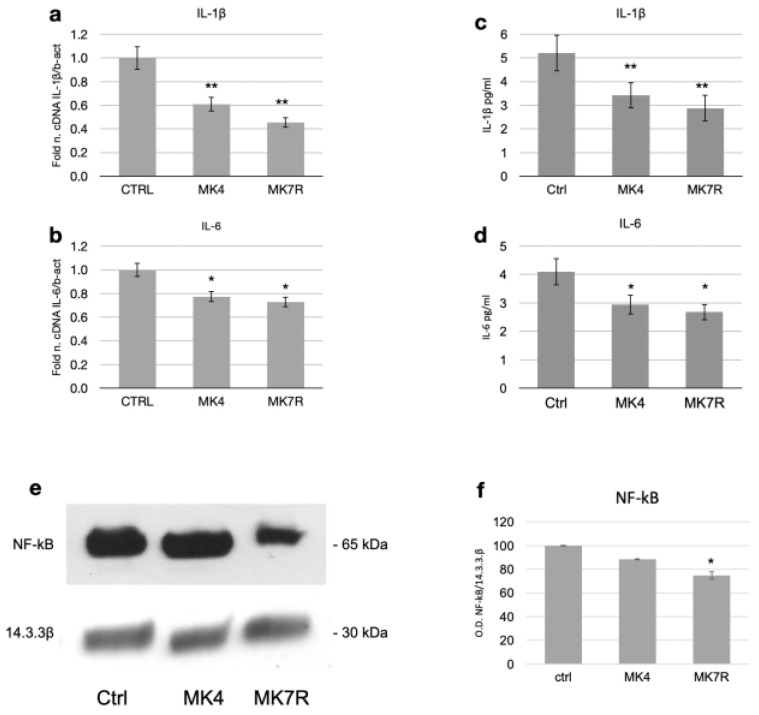
Expression of factors involved in neuroinflammatory molecular processes in SK-N-BE cells treated for 48 (mRNA) or 72 h (proteins) with 50 µM MK4 or MK7R. mRNA expression of (**a**) IL-1β and (**b**) IL-6. Bar plots show the relative amounts of the target genes normalized to β-actin internal reference obtained using real-time PCR on the *y*-axis. (**c**,**d**) ELISA assay for the detection of circulating pro-inflammatory cytokines. Bar plots show the concentration expressed in pg/mL of (**c**) IL-1β and (**d**) IL-6. Representative Western blot images (**e**) of NF-kB and 14.3.3β proteins. (**f**) Densitometric analysis of protein bands obtained in at least 3 independent experiments using the 14.3.3β band as internal reference. Values are expressed as the percent of the control. Abbreviations: MK4, menaquinone 4; MK7R, reduced menaquinone 7. * *p* < 0.05, ** *p* < 0.01 vs. ctrl; *N* = 3.

**Figure 5 cells-13-00058-f005:**
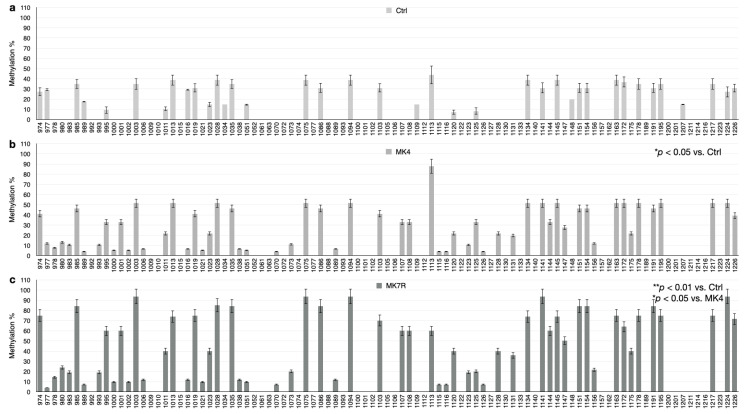
*PSEN1* promoter CpG and non-CpG methylation patterns in SK-N-BE cells treated for 48 h with 50 µM MK4 or MK7R. Bar plots in (**a**) ctrl, (**b**) MK4 and (**c**) MK7R show methylation percentage (±standard deviation, *y*-axis) of each cytosine; labels on *x*-axis indicate the cytosine position on the reference sequence from the 5′ (**left**) to the 3′ (**right**) of the promoter, i.e., with the region proximal to the transcription start site (**right**). The level of statistical significance using contingency table analysis is shown on the right side of the bar plots for MK4 vs. ctrl (**b**) and MK7R vs. ctrl and MK4 (**c**). Abbreviations: MK4, menaquinone 4; MK7R, reduced menaquinone 7. * *p* < 0.05, ** *p* < 0.01 vs. ctrl; *N* = 3.

**Figure 6 cells-13-00058-f006:**
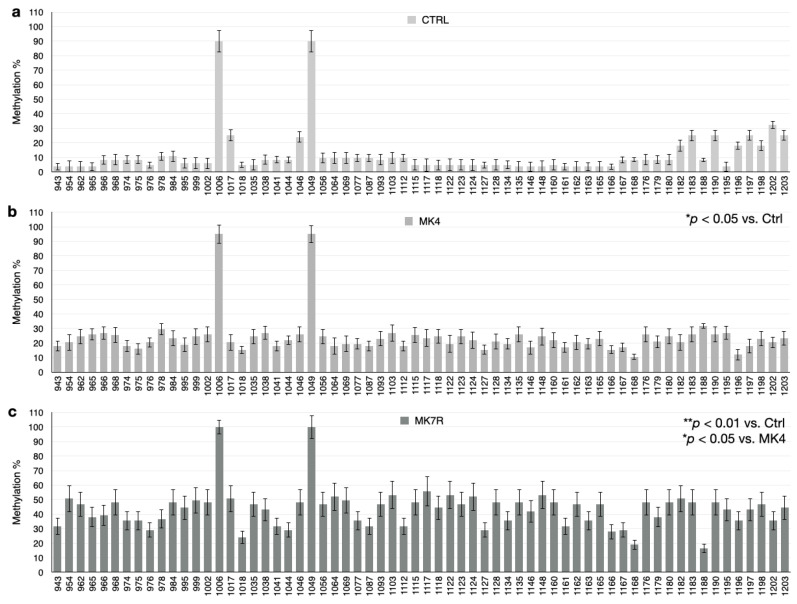
*IL-1β* promoter CpG and non-CpG methylation patterns in SK-N-BE cells treated for 48 h with 50 µM MK4 or MK7R. Bar plots in (**a**) ctrl, (**b**) MK4 and (**c**) MK7R show methylation percentage (±standard deviation, *y*-axis) of each cytosine; labels on *x*-axis indicate the cytosine position on the reference sequence from the 5′ (**left**) to the 3′ (**right**) of the promoter, i.e., with the region proximal to the transcription start site (**right**). The level of statistical significance using contingency table analysis is shown on the right side of the bar plots for MK4 vs. ctrl (**b**) and MK7R vs. ctrl and MK4 (**c**). Abbreviations: MK4, menaquinone 4; MK7R, reduced menaquinone 7. * *p* < 0.05, ** *p* < 0.01 vs. ctrl; *N* = 3.

**Figure 7 cells-13-00058-f007:**
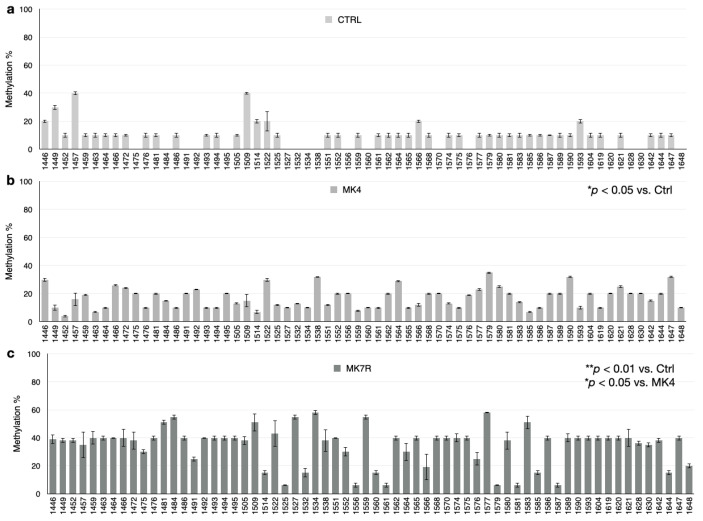
*IL-6* promoter CpG and non-CpG methylation patterns in SK-N-BE cells treated for 48 h with 50 µM MK4 or MK7R. Bar plots in (**a**) ctrl, (**b**) MK4 and (**c**) MK7R show methylation percentage (±standard deviation, *y*-axis) of each cytosine; labels on *x*-axis indicate the cytosine position on the reference sequence from the 5′ (**left**) to the 3′ (**right**) of the promoter, i.e., with the region proximal to the transcription start site (**right**). The level of statistical significance using contingency table analysis is shown on the right side of the bar plots for MK4 vs. ctrl (**b**) and MK7R vs. ctrl and MK4 (**c**). Abbreviations: MK4, menaquinone 4; MK7R, reduced menaquinone 7. * *p* < 0.05, ** *p* < 0.01 vs. ctrl; *N* = 3.

**Figure 8 cells-13-00058-f008:**
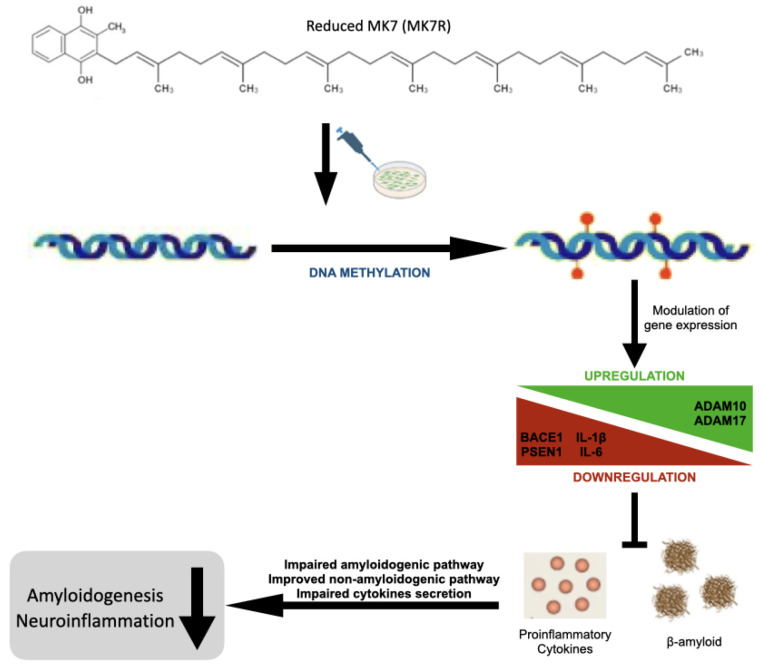
Schematic graphical summary of the results.

**Table 1 cells-13-00058-t001:** Primers used in real-time PCR assays.

Gene Name	Primer	Primer Sequence (5′-3′)	Primer Length (bp)	Amplicon Size (bp)
β-actin	Forward	CAACCGCGAGAAGATGACC	19	94
	Reverse	AGAGGCGTACAGGGATAGCA	20	
PSEN 1	Forward	GGTCGTGGCTACCATTAAGTC	21	94
	Reverse	GCCCACAGTCTCGGTATCTT	20	
BACE 1	Forward	AACGAATTGGCTTTGCTGTC	20	102
	Reverse	AGCCACAGTCTTCCATGTCC	20	
APP	Forward	GAACTACATCACCGCTCTGC	20	77
	Reverse	CGCGGACATACTTCTTTAGC	20	
ADAM10	Forward	TGCCATGTATGCTGTATGAAGA	22	109
	Reverse	ATCCAGGTTGCAGGGTGAT	19	
ADAM17	Forward	TCAAGAATGTTTCACGTTTGC	21	81
	Reverse	ACCCTTTTGGGAGCAACTCT	20	
IL-1β	Forward	TCCCCAGCCCTTTTGTTGA	19	332
	Reverse	TTAGAACCAAATGTGGCCGTG	21	
IL-6	Forward	GGCACTGGCAGAAAACAACC	20	257
	Reverse	GCAAGTCTCCTCATTGAATCC	21	

**Table 2 cells-13-00058-t002:** Antibodies used in WB assays.

Protein	Antibody		Manufacturer	Band Size
APP	MAB348	Monoclonal	Chemicon International, Temecula, CA, USA	110 kDa
PSEN1	MAB5232	Monoclonal	Chemicon International, Temecula, CA, USA	18–20 kDa
BACE 1	sc-10055	Polyclonal	Santa Cruz Byotechnology, Santa Cruz, CA, USA	70 kDa
ADAM10	AB19026	Polyclonal	Chemicon International, Temecula, CA, USA	60–85 kDa
ADAM17	sc-6416	Polyclonal	Santa Cruz Byotechnology, Santa Cruz, CA, USA	85 kDa
sAPPα	11088	Monoclonal	IBL, Gunma, Japan	80–100 kDa
NF-kB	D14E12	Monoclonal	Cell Signaling Technology	65 kDa
14-3-3β	sc-629	Polyclonal	Santa Cruz Byotechnology, Santa Cruz, CA, USA	30 kDa

## Data Availability

The datasets used and/or analyzed during the current study are available from the corresponding author upon reasonable request.

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
