# Peer review of "Amyloidogenic and Neuroinflammatory Molecular Pathways Are Contrasted Using Menaquinone 4 (MK4) and Reduced Menaquinone 7 (MK7R) in Association with Increased DNA Methylation in SK-N-BE Neuroblastoma Cell Line"

_cells, 2023, doi:10.3390/cells13010058_

Round 1

Reviewer 1 Report

Comments and Suggestions for Authors

The paper by Orticello et al. is an intriguing contribution aimed at elucidating the role of two vitamin K2 vitamers (MK4 and MK7R) in amyloidogenic and neuroinflammatory pathways. The applied methods and all the results are presented clearly, and they appear to be sound.

Below please find some points requiring further elaboration or revision by the authors:

1. Some relevant and very recent references are missing. I suggest including them since they are strictly related to the importance and the delicate role played by vitamins in neurodegeneration

- Could Vitamins Have a Positive Impact on the Treatment of Parkinson's Disease? Sandeep, Sahu MR, Rani L, Kharat AS, Mondal AC. Brain Sci. 2023 Feb 6;13(2):272. doi: 10.3390/brainsci13020272. PMID: 36831815 Free PMC article. Review.

 - Bioinorganic Chemistry of Micronutrients Related to Alzheimer's and Parkinson's Diseases. Kola A, Nencioni F, Valensin D. Molecules. 2023 Jul 17;28(14):5467. doi: 10.3390/molecules28145467.

- Vitamin K2 protects against Aβ42-induced neurotoxicity by activating autophagy and improving mitochondrial function in Drosophila. Lin X, Wen X, Wei Z, Guo K, Shi F, Huang T, Wang W, Zheng J. Neuroreport. 2021 Apr 7;32(6):431-437. doi: 10.1097/WNR.0000000000001599.

- Protective effects of vitamin K2 on 6-OHDA-induced apoptosis in PC12 cells through modulation bax and caspase-3 activation. Ramazani E, Fereidoni M, Tayarani-Najaran Z. Mol Biol Rep. 2019 Dec;46(6):5777-5783. doi: 10.1007/s11033-019-05011-2. Epub 2019 Aug 7.

 -The Impact of Vitamin E and Other Fat-Soluble Vitamins on Alzheimer´s Disease. Grimm MO, Mett J, Hartmann T. Int J Mol Sci. 2016 Oct 26;17(11):1785. doi: 10.3390/ijms17111785.

2. At pag 2, line 75, the authors have mentioned at Figure 1. Unfortunately Figure 1 is not included in the manuscript. 

3. At pag.3 lines 133 the authors wrote "For this study we used MK 4, MK7, MK7R and diacetylated-MK7". However, all the analysis reported in the manuscript are just referred to MK4 and MK7R. The authors should better clarify this point.

4. At pag 3, line 133 substitute MK 4 with MK4. 

5. At pag., line 138 Supplementary Figure 1 is not consistent with MK4 and MK7R compounds as indicated. 

6. The authors need to better explain the choice of using MK4 and MK7R vitamers instead for example of MK7 since it is not clear enough.

8. Pag.3 lines 141-149, did the authors use MK7R obtained by the vendor, or they prepared it from MK7. Please clarify this point which is not clear.

9. Original WB images shown in Supplementary Figure 4 are difficult to comprehend.

10. Pag. 9 lines 339, 340, 343 please substitute Fig. with Figure.

11. The resolution of Figure 8 is very poor.

12. Pag 16, line 536, the text description of Figure S4 and S5 is reverted.

Author Response

Reviewer 1

The paper by Orticello et al. is an intriguing contribution aimed at elucidating the role of two vitamin K2 vitamers (MK4 and MK7R) in amyloidogenic and neuroinflammatory pathways. The applied methods and all the results are presented clearly, and they appear to be sound.

R: We are sincerely grateful to the reviewer for the very positive comment and evaluation.

  1. Some relevant and very recent references are missing. I suggest including them since they are strictly related to the importance and the delicate role played by vitamins in neurodegeneration

R: We are grateful to the reviewer for having suggested these references that update our literature analysis. These citations have been added in the Discussion

  1. At pag 2, line 75, the authors have mentioned at Figure 1. Unfortunately Figure 1 is not included in the manuscript. 

R: We are grateful to the reviewer for having noticed the lacking of the figure, probably occurred during the formatting of the manuscript. The figure has been now inserted again.

  1. At pag.3 lines 133 the authors wrote "For this study we used MK 4, MK7, MK7R and diacetylated-MK7". However, all the analysis reported in the manuscript are just referred to MK4 and MK7R. The authors should better clarify this point.

R: We are grateful for the comment that highlight a point of scarce clarity. (See also our reply to the comment #6, which deals with the same aspect. We now explained this point at the end of the introduction.

  1. At pag 3, line 133 substitute MK 4 with MK4. 

R: The typo has been corrected.

  1. At pag., line 138 Supplementary Figure 1 is not consistent with MK4 and MK7R compounds as indicated. 

R: we are sorry for the mistake: the figure to be indicated here should have been “Figure 1”, not “Supplementary Figure 1”.

  1. The authors need to better explain the choice of using MK4 and MK7R vitamers instead for example of MK7 since it is not clear enough.

MK7 and diacethylated MK7 have been initially considered but then abandoned, after the initial assays, due to their poor solubility in the culture medium, since this could have caused poor reliability and inaccurate control of the real concentration. This aspect is now better described at the end of the Introduction

  1. Pag.3 lines 141-149, did the authors use MK7R obtained by the vendor, or they prepared it from MK7. Please clarify this point which is not clear.

The molecules were prepared by Gnosis by Lesaffre and shipped to our lab. This point has been now clarified.

  1. Original WB images shown in Supplementary Figure 4 are difficult to comprehend.

We understand the difficulties. The figure has been generated as it is to accommodate the journal’s request to produce the original, uncrossed and unmodified scan of the wb films used for the semiquantitative analysis (I.e in triplicate). Each film reports (signed by pen, as taken from the development): the protein name (below or above the image); the experimental condition above the image (were not indicated, the bands are to be intended in the same order Ctrl, MK4, MK7); the molecular weight of the band (in kDa) left or right the specific bands; the exposure time in seconds; the pen signs delimitating the membrane; the A or B letter when we performed multiple repetitions of the samples in the same gel.

We hope this clarification helps to better read the image.

  1. Pag. 9 lines 339, 340, 343 please substitute Fig. with Figure.

R: We changed the text according to the request

  1. The resolution of Figure 8 is very poor.

R: In agreement with reviewer’s observation, we see that the pdf of the manuscript on the journal’s website includes a low-resolution figure. On the other hand, the docx file has the correct and high resolution figure version. This is probably due to a problem occurred during the export by the journal and we will highlight this point with the editorial office at the resubmission.

  1. Pag 16, line 536, the text description of Figure S4 and S5 is reverted.

R: We are grateful for having noticed the reversion. It is now corrected

Reviewer 2 Report

Comments and Suggestions for Authors

This article explores the potential of menaquinone-4 (MK4) and reduced menaquinone-7 (MK7R) in contrast to amyloidogenic and neuroinflammatory molecular pathways. The study was conducted on SK-N-BE human neuroblastoma cells, which were treated with MK4 and MK7R for 72 hours. The results showed that MK4 and MK7R were able to reduce amyloidogenesis and neuroinflammation, while also increasing DNA methylation. The authors suggest that Vitamin K2 may play a role in gene modulation and epigenetic regulation, and that further research is needed to fully understand its potential in preventing neurodegenerative diseases.

The authors offer a comprehensive introduction, delineating the role of vitamin K beyond coagulation and its potential implications in inflammation and age-related diseases. This contextual information establishes the study's significance. The choice of the SK-N-BE neuroblastoma cell line as a model for analyzing neuroinflammatory and neurodegenerative pathways is good. The incorporation of both MK4 and the newly assessed MK7R as vitamer compounds permits an exhaustive analysis.

The study investigates the expression of pivotal factors in amyloidogenesis and neuroinflammation, yielding valuable insights into MK4 and MK7R's potential impact on these pathways. The identified correlation between hypermethylation and downregulation of specific genes adds a nuanced layer of understanding, suggesting a potential epigenetic mechanism for MK7R.

The study provides valuable insights into the divergent effects of MK4 and MK7R on amyloidogenic and neuroinflammatory pathways.

Minor concerns:

1) In the employed cellular model, treatment with MK4 and MK7R is linked to decreased expression of BACE1 and PSEN1 alongside hyper-expression of ADAM10 and ADAM17. However, methylation profiles were specifically defined for PSEN1, IL1, and IL6. Were the authors able to substantiate the methylation status of the expression regulatory regions of BACE1, ADAM10, and ADAM17 genes?

2) The study notes a correlation between hypermethylation and downregulation of specific genes implicated in amyloidogenesis and neuroinflammation. While this correlation is valuable, it does not establish a causal relationship. Further experiments, such as functional studies or manipulation of DNA methylation levels, would be necessary to determine the direct impact of DNA methylation on gene expression. Did the authors consider the possibility of using DNMT inhibitors like the cytosine analogue 5'-azacytidine (5'-aza) to confirm whether this treatment validates that the expression of PSEN1, IL-1β, and IL-6 genes is epigenetically regulated?

Author Response

Response to Reviewer 2

[…] The study provides valuable insights into the divergent effects of MK4 and MK7R on amyloidogenic and neuroinflammatory pathways.

R: The authors are sincerely grateful to the reviewer for the positive comments.

1) In the employed cellular model, treatment with MK4 and MK7R is linked to decreased expression of BACE1 and PSEN1 alongside hyper-expression of ADAM10 and ADAM17. However, methylation profiles were specifically defined for PSEN1, IL1, and IL6. Were the authors able to substantiate the methylation status of the expression regulatory regions of BACE1, ADAM10, and ADAM17 genes?

R: The comment of the reviewer is helpful since it evidence that this point wasn’t sufficiently clear: in previous works we demonstrated that BACE1, ADAM10 and ADAM17 expression, although modulated in association with hypo- and hyper-methylating treatments, was not directly dependent on altered DNA methylation. For this reason we did not explored the methylation of their promoters in the present (and in other) works, although we are actively looking for “mediators” of the epigenetic effect in other research projects.

We added a sentence in the Discussion to better clarify this point.

2) The study notes a correlation between hypermethylation and downregulation of specific genes implicated in amyloidogenesis and neuroinflammation. While this correlation is valuable, it does not establish a causal relationship. Further experiments, such as functional studies or manipulation of DNA methylation levels, would be necessary to determine the direct impact of DNA methylation on gene expression. Did the authors consider the possibility of using DNMT inhibitors like the cytosine analogue 5'-azacytidine (5'-aza) to confirm whether this treatment validates that the expression of PSEN1, IL-1β, and IL-6 genes is epigenetically regulated?

R: We sincerely appreciate the reviewer’s suggestion. We did studied in deep and already published (some of the papers are cited in the present manuscript) the causal association between methylation and gene expression by using hypomethylating (B vitamin deficiency) and hypermethylating (S-adenosylmethionine supplementation) conditions. Of course, the mechanistic connection between vitamin K and methylation is not yet clear, as disclosed in the Discussion. Regarding the use of 5-aza, although this is a wide used protocol to induce hypoomethylation, in our studies we usually prefer to use B vitamin deficiency since it exerts its effect through the one-carbon metabolism, which is the biochemical pathway regulating the methyl groups availability. Therefore, the B deficiency results in a more “physiologic” intervention respect to 5-aza, that incorporates into DNA and cannot be methylated, independently on the “normal” regulation of methylation by the cell.